behaviour/ecology

migration, non-breeding, moult, stable isotopes, Procellariiformes, Southern Ocean

**Author for correspondence:**
Aymeric Fromant
e-mail: afromant@deakin.edu.au

# Temporal and spatial differences in the post-breeding behaviour of a ubiquitous Southern Hemisphere seabird, the common diving petrel

Aymeric Fromant[1,2], Charles-André Bost[2],
Paco Bustamante[3,4], Alice Carravieri[3], Yves Cherel[2],
Karine Delord[2], Yonina H. Eizenberg[1], Colin M. Miskelly[5]
and John P. Y. Arnould[1]

[1]School of Life and Environmental Sciences, Deakin University, 221 Burwood Highway, Burwood, Victoria 3125, Australia
[2]Centre d'Etudes Biologiques de Chizé (CEBC), UMR 7372 CNRS—La Rochelle Université, 79360 Villiers en Bois, France
[3]Littoral Environnement et Sociétés (LIENSs), UMR 7266 CNRS—La Rochelle Université, 2 rue Olympe de Gouges, 17000 La Rochelle, France
[4]Institut Universitaire de France (IUF), 1 rue Descartes, 75005 Paris, France
[5]Museum of New Zealand Te Papa Tongarewa, PO Box 467, Wellington 6140, New Zealand

AF, 0000-0002-3024-7659; C-AB, 0000-0002-5259-9073;
PB, 0000-0003-3877-9390; AC, 0000-0002-7740-843X;
YC, 0000-0001-9469-9489; KD, 0000-0001-6720-951X;
YHE, 0000-0001-6080-5650; JPYA, 0000-0003-1124-9330

The non-breeding period plays a major role in seabird survival and population dynamics. However, our understanding of the migratory behaviour, moulting and feeding strategies of non-breeding seabirds is still very limited, especially for small-sized species. The present study investigated the post-breeding behaviour of three distant populations (Kerguelen Archipelago, southeastern Australia, New Zealand) of the common diving petrel (CDP) (*Pelecanoides urinatrix*), an abundant, widely distributed zooplanktivorous seabird breeding throughout the southern Atlantic, Indian and Pacific oceans. The timing, geographical destination and activity pattern of birds were quantified through geolocator deployments during the post-breeding migration, while moult pattern of body feathers was investigated using stable isotope analysis. Despite the high

energetic cost of flapping flight, all the individuals quickly travelled long distances (greater than approx. 2500 km) after the end of the breeding season, targeting oceanic frontal systems. The three populations, however, clearly diverged spatially (migration pathways and destinations), and temporally (timing and duration) in their post-breeding movements, as well as in their period of moult. Philopatry to distantly separated breeding grounds, different breeding phenologies and distinct post-breeding destinations suggest that the CDP populations have a high potential for isolation, and hence, speciation. These results contribute to improving knowledge of ecological divergence and evolution between populations, and inform the challenges of conserving migratory species.

# 1. Introduction

In order to adapt to seasonal environmental changes, many animal species migrate during the non-breeding period [1]. Post-breeding migration is particularly common in seabirds [2], some species travelling long distances between breeding and non-breeding habitats [3,4]. The post-breeding period is particularly important for plumage replacement for a wide range of seabirds [5]. This period can, therefore, be energetically challenging [6,7] and the divergent environmental conditions experienced by populations throughout their migration and at the moulting areas may differentially influence demographic processes [8–10]. Different migratory and moult strategies within a species may lead to long-term population divergences, and carry important ecological, evolutionary and conservation implications [11].

Distinct populations within a species can have substantial demographic and ecological differences [12–14]. As the response of one particular population to environmental changes may not be representative of the entire species, it is important to obtain information from multiple populations [15,16]. However, only a few coordinated studies have investigated multiple separate populations of seabirds during the non-breeding period [9]. This is particularly true in small-sized seabirds, even though they are more likely to be vulnerable to anthropogenic threats than larger species [17]. In conservation biology, failing to consider population-specific ecology during the non-breeding period may lead to a misunderstanding of the potential effects of local threats or environmental changes on population dynamics of small seabird species [13]. Therefore, knowledge of the distribution and at-sea activity of multiple populations is required in order to determine the inter-population diversity during this critical period [15,17,18].

Stable isotope analysis of feathers provides a robust method to study the non-breeding feeding ecology of seabirds [19,20] and investigate inter-population variability [21]. Values of $\delta^{13}C$ in feathers provide information on the feeding habitat occupied during moult [22] and $\delta^{15}N$ values provide highly valuable indications on the trophic niche [23]. This approach is nonetheless unable to provide precise temporal and spatial scales of the migratory behaviour of seabirds.

Owing to the high-energy cost of carrying telemetry devices, there is a trade-off between the size of animal-borne transmitters and their deployment duration such that their application in seabirds has been constrained to species with larger body size and/or for a relative short period [24]. Daylight sensing geolocation provides a lightweight alternative for long-term deployments (geolocation sensing data loggers, GLS) and is increasingly being used to investigate the at-sea distribution and activity of small procellariiform species during the breeding [25,26] and non-breeding periods [27–29].

Non-breeding ecology has been investigated in a wide range of large seabirds in the Southern Ocean (e.g. albatrosses, shearwaters and penguins; [30]). While some studies have recently investigated the distribution of multiple populations of small petrels during this period, most have focused on a small proportion of populations and/or limited spatial range of the species [28,31]. The common diving petrel (CDP, *Pelecanoides urinatrix*) is a small procellariiform seabird (110–160 g) feeding on macrozooplankton [32–34]. This species breeds on numerous islands along a wide latitudinal gradient (35°–55° S) throughout the Atlantic, Indian and Pacific oceans, with an estimated 8–10 million breeding pairs (figure 1; [5]), though substantial differences in breeding phenology are evident among populations (figure 1). Recent tracking studies suggest that CDP head to the subantarctic region during the post-breeding migration [25,40]. This observation is well supported by isotopic analysis of feathers of individuals from the Kerguelen Archipelago and South Georgia [21,25]. However, the distribution of CDP during the non-breeding season remains unknown for the majority of breeding sites, particularly in the Indian Ocean sector of the Southern Ocean which hosts large populations [36,37,41,42]. Furthermore, it is uncertain if these different populations share the same ecology (spatial, temporal and trophic segregation) and whether they consistently migrate to the same regions.

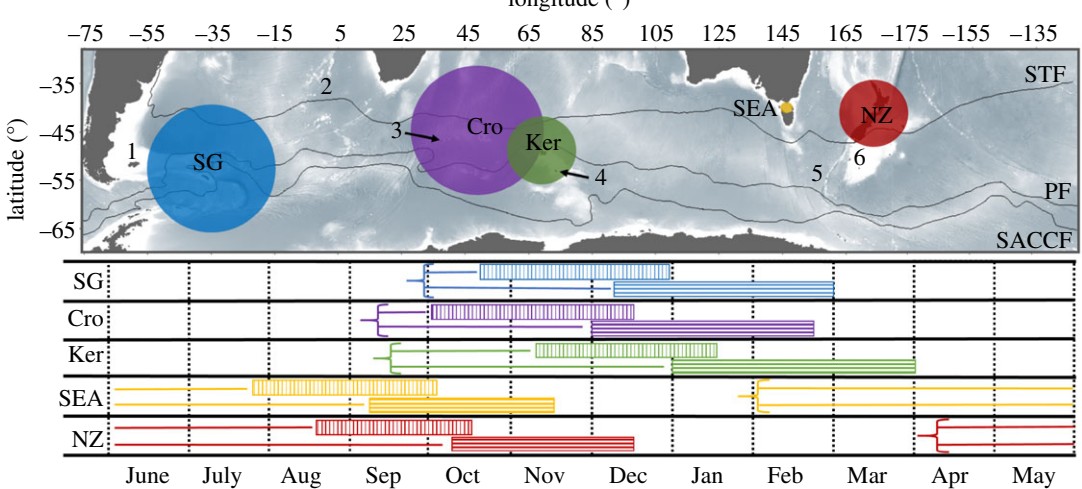

**Figure 1.** Upper panel: population estimates of CDPs (*P. urinatrix*) throughout its circumpolar distribution. The circles show the proportional sizes of the main populations: South Georgia (SG; blue circle; 3.8 million breeding pairs [35]); Crozet (Cro; purple circle; 3–4 million breeding pairs [36]); Kerguelen (Ker; green circle; 0.5–1 million breeding pairs [37]); southeastern Australia (SEA; orange circle; 0.1–0.2 million breeding pairs [38]; New Zealand (NZ; red circle; 0.9 million breeding pairs [39]). The numbers 1–6 show small isolated populations (less than 50 000 breeding pairs): 1, Falklands/Malvinas islands; 2, Gough/Tristan da Cuna Islands; 3, Prince Edwards Islands; 4, Heard/McDonald Islands; 5, Macquarie Island; 6, New Zealand subantarctic Islands. The black lines represent the approximate location of the Subtropical Front (STF), Polar Front (PF) and Southern Antarctic Circumpolar Current Front (SACCF). Lower panel: phenology of CDPs from South Georgia (SG), Crozet (Cro), Kerguelen (Ker), southeastern Australia (SEA) and New Zealand (NZ). Blocks with vertical lines correspond to incubation and horizontal lines to chick-rearing period. Arrows followed by horizontal lines show the period when the species is present at a specific breeding site. For Ker, SEA and NZ, the data correspond to the colonies surveyed in the present study in 2017–2018.

Similarly, there is very limited information about the critical moulting period, and how breeding phenology could influence the subsequent post-breeding activity patterns.

The present study investigated the post-breeding behaviour of CDP from three distantly separated breeding sites from two oceanic basins in the Southern Hemisphere, by combining information from GLS and isotopic analyses. The specific aims were to: (i) identify the spatial distribution of post-breeding individuals; (ii) describe the timing and duration of their post-breeding movements and moult; and (iii) evaluate the potential differences among the focal populations, and discuss their implication for further conservation biology studies. Additionally, by comparing our results with those of CDP populations from three other localities in the southern Atlantic and Pacific oceans [25,40], we explored the extent to which the timing and location of post-breeding movements contribute to the ecological and evolutionary divergence of distinct populations of a ubiquitous seabird species.

## 2. Methods

### 2.1. Study site, study species and animal instrumentation

The study was conducted at three field sites representing separate populations: Ile Mayes (Kerguelen Archipelago, southern Indian Ocean, 49°28′ S 69°57′ E, hereafter referred to as Kerguelen); Kanowna Island (southeastern Australia, 39°10′ S 148°16′ E) and Mana Island (New Zealand, 41°06′ S 174°46′ E). In order to avoid potential bias owing to inter-annual variations, individuals from all three colonies were sampled during the same non-breeding period (November 2017–September 2018). Additional data were collected for Kerguelen (in 2015–2016 and 2016–2017) and Kanowna Island (in 2016–2017 and 2018–2019). Kerguelen is located in subantarctic waters *sensu lato* (between the Polar Front and the Subtropical Front), while both Kanowna and Mana Islands are located farther north, in the temperate subtropical region.

In the present article, migration is defined as the period during which there is an annual to-and-from movement of populations between the breeding site and a migration area [1]. The post-breeding period is

defined as the period between the last burrow attendance at the end of the breeding season and the first return in a burrow the following season. The non-breeding period is the period between two successive breeding cycles. It includes the post-breeding migration away from breeding sites, followed by a more or less prolonged period in the vicinity of breeding sites before initiating the next breeding season.

To study the at-sea distribution of individuals from the three populations during the non-breeding period, adult birds were equipped with leg-mounted GLS (Migrate Technology, model C65). The mass of the device attached to a plastic or metal ring with a cable tie was less than 1.5 g, corresponding to on average $1.07 \pm 0.1\%$ of body mass (117–175 g). Previous use of such devices on small seabird species have shown limited impact on the feeding ecology or future breeding [27,43]. Breeding individuals were equipped at the end of the breeding season and were recaptured during the following breeding season for removal of the device. Sex was determined by DNA analysis of a small blood sample (0.1 ml) collected from the brachial vein for the individuals from Kerguelen (Laboratoire Analyses Biologiques, CEBC, France), and of a body feather for those from Kanowna Island (DNA Solutions, Australia). The sex of individuals at Mana Island was determined from their sex-specific calls [5,44].

Information on moult pattern in Pelecanoididae species is very limited, largely because individuals can be at sea during this stage [5]. Similar to alcids, the majority of diving petrel species seem to undergo an annual almost synchronous wing moult of flight feathers during the non-breeding period [45,46]. For CDP, in southeastern Australia and New Zealand, adults are thought to start wing moult at sea after chicks have fledged [5]. In South Georgia, Kerguelen and Heard Island, moult of primaries seems to be initiated just before the end of the chick-rearing period [5,47], although sources are discordant [48]. However, the accuracy of these records is complicated by the lack of information on the breeding status of the observed individuals. In the present study, the timing of wing moult was inferred from information provided by the GLS on the time spent on the sea surface. Because flight feather renewal directly affects flying ability [46,49,50], in particular for species with high wing loading (body mass/wing area) such as diving petrels [51], a peak in time spent on the water is likely to reflect the wing moult of flight feathers.

For body feathers, it is likely to be a protracted process initiated during the post-breeding period, similar to other species of small petrels [52,53]. The continuous moult of body feathers throughout the non-breeding period allows the birds to progressively renew their plumage while maintaining its vital role in thermoregulation [54]. Stable isotope ratios in body feathers can be a proxy of the location and trophic level of the individuals when they were synthesized [19,20]. Therefore, to investigate the isotopic niche and the moult of body feathers of individuals throughout the non-breeding period, four contour feathers were collected from the middle and lower back of each tracked individual at recapture. Additional samples were collected on adult individuals that were not tracked in order to increase the sample size and to assess temporal variation (3 years of data for both Kerguelen and Kanowna). Handling time at deployment (measurements, banding and GLS attachment) and recapture (device removal, weighing, blood and feather sampling) were minimized and took on average less than 5 min.

## 2.2. Data processing and analyses

The GLS measured light intensity every minute and recorded the maximum value for each 5 min interval. The determination of twilights (dawn and dusk), linked to a time base, enabled longitude (timing of local midday and midnight) and latitude (duration of day and night) to be estimated, providing two positions per day with an average of $186 \pm 114$ km (mean error $\pm$ s.d.; [55]). Processing and calculations were conducted using the *GeoLight* package in the R statistical environment [56,57]. As latitude estimations around the equinoxes are unreliable, data for two weeks before and after the equinoxes were excluded before spatial analysis was conducted [58]. Additionally, latitude or longitude estimates that were clearly inaccurate (unrealistic speed greater than 1500 km d$^{-1}$, trajectory or spikes) were removed [31]. When outward and inward movements were occurring during the equinox periods (March–April and September–October), the timing of arrival and departure from the breeding colonies was determined from longitudinal directional movements using the raw data [31,50].

Filtered locations were used to generate kernel utilization distribution (UD) estimates with a smoothing parameter ($h$) of 1.8 (corresponding to a search radius of approx. 200 km) and $1° \times 1°$ grid cell size (based on the mean accuracy of the device). For each population and year, the 50% (core foraging area) and 95% (home range) kernel UD contours were obtained [59]. The core area was used to estimate the centroid position (mean latitude and longitude) during the post-breeding period for each individual. Spatial analyses were performed using the *adehabitatHR* R package [60].

The period of maximum proportion of time spent on the water (wet–dry sensor being wet greater than 90% per day; [40]) was used to determine the moulting period of flight feathers for each individual [50]. This was recorded differently at the three study sites. For Kerguelen and Kanowna Island, wet–dry data were sampled every 30 s with the number of samples wet and maximum conductivity recorded every 4 h. At Mana Island, wet–dry data were sampled every 30 s with the number of wet samples and maximum conductivity recorded every 10 min.

The dates of last and first burrow attendance were determined by combining information on activity (wet–dry: 100% dry for a period greater than 4 h), temperature (for Kerguelen and Kanowna Island only, sampled every 5 min with maximum recorded every 4 h) and movement data (presence of the bird in the vicinity of the breeding region). These data were then used to estimate the duration and the total distance travelled during the post-breeding migration.

For stable isotope analyses, feathers were cleaned of surface lipids and contaminants using a 2 : 1 chloroform : methanol solution in a ultrasonic bath, followed by two successive methanol rinses and air dried 24 h at 50°C. Each feather was then cut to produce a fine powder for homogenization before carbon and nitrogen isotope ratio determination using a continuous flow mass spectrometer (Delta V Plus or Delta V Advantage both with a Conflo IV interface, Thermo Scientific, Bremen, Germany) coupled to an elemental analyser (Flash 2000 or Flash EA 1112, Thermo Scientific, Milan, Italy) at the LIENSs laboratory (La Rochelle Université, France). Stable isotope values were expressed in conventional notation ($\delta X = [(R_{sample}/R_{standard}) - 1]$) where $X$ is $^{13}C$ or $^{15}N$ and $R$ represents the corresponding ratio $^{15}N/^{14}N$ or $^{13}C/^{12}C$. $R_{standard}$ values were based on Vienna Pee Dee Belemnite for $^{13}C$, and atmospheric nitrogen ($N_2$) for $^{15}N$. Replicates of internal laboratory standards (Caffeine USGS-61 and USGS-62) indicate measurement errors less than 0.10 ‰ for $\delta^{13}C$ and 0.15 ‰ for $\delta^{15}N$.

As the Southern Ocean is marked by a latitudinal gradient of $\delta^{13}C$ and $\delta^{15}N$ (Jaeger *et al.* [61]), low ($\delta^{13}C < -19.5$ ‰ and $\delta^{15}N < 9.9$ ‰) and high isotopic values ($\delta^{13}C > -19.5$ ‰ and $\delta^{15}N > 9.9$ ‰) in body feathers were interpreted as corresponding to feathers moulted in Antarctic/subantarctic and subtropical/neritic waters, respectively [21,61].

All spatial and statistical analyses were conducted in the R statistical environment 3.6.1 [57]. To investigate among-population, inter-annual and sex-related variations on post-breeding migration parameters (departure and return dates, migration duration, total distance travelled and maximum range), general linear models (GLMs) were fitted using the *lme4* package [62]. For all models, a Gaussian family was selected (error structure approached the normal distribution), all combinations of variables were then tested and ranked based on their Akaike's information criterion (AIC), and the global models were checked to ensure the normality and homoscedasticity of the residuals [63] before further statistical tests. Inter-population differences were quantified using analyses of variance (ANOVA or Welch's ANOVA), and *post hoc* tests were conducted using *t*-tests (parametric), or Kruskal–Wallis and Mann–Whitney *U*-tests (non-parametric) depending on the data distributions. Before these analyses, data were checked for normality (Shapiro–Wilk test) and equality of variances (Levene test). Estimates are presented as means ± s.e. (unless specified).

# 3. Results

In total, 21 GLS (nine in 2015–2016 and 12 in 2017–2018) were deployed on CDPs from Kerguelen, 41 (20 in 2017–2018 and 21 in 2017–2019) from Kanowna Island and 10 (in 2017–2018) from Mana Island. Recapture rate was 68% (70% in 2015–2016 and 66% in 2017–2018) at Kerguelen, 31% (25% in 2017–2018 and 38% in 2018–2019) at Kanowna Island and 40% at Mana Island. At Kerguelen, two GLS (one in 2015–2016 and one in 2017–2018) were found in brown skua (*Catharacta antarctica lonnbergi*) pellets near the colony before the end of the breeding season. Additionally, owing to a malfunction, all the GLS deployed at Kerguelen in 2015–2016 stopped recording three to six months after deployment (just before or soon after departure from the colony) and, therefore, data from these individuals were excluded from further statistical analysis. Because there was no significant difference between the sexes in at-sea distribution, trip parameters, and stable isotope values (electronic supplementary material, table S1), data were pooled across sexes in all subsequent statistical analyses.

## 3.1. Post-breeding migration patterns and at-sea activity

All individuals migrated predominantly to the vicinity of the Polar Front, while birds from Kerguelen moved farther south near the Southern Antarctic Circumpolar Current Front (figure 2). In 2017–2018

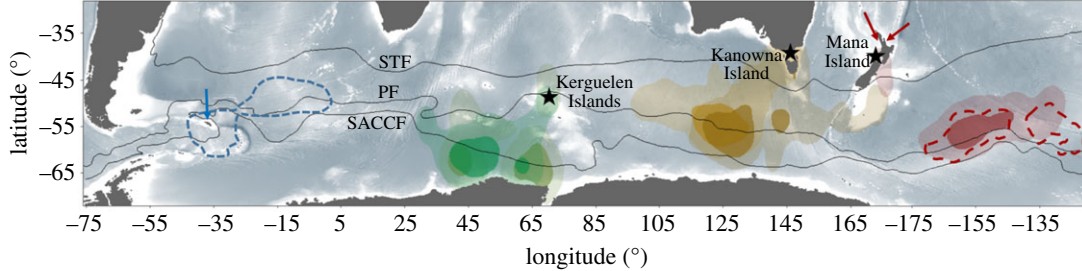

**Figure 2.** Kernel density estimation derived from GLS locations during the post-breeding migration of CDPs (*P. urinatrix*) from Kerguelen (light green in 2015–2016, *n* = 5; and dark green in 2017–2018, *n* = 7), Kanowna (orange in 2017–2018, *n* = 5; and brown in 2018–2019, *n* = 8) and Mana Islands (red in 2017–2018, *n* = 4). Solid colour areas show the 50% (core area) and faint colour the 95% (home range) of the kernel UD. Dotted lines represent the estimated post-breeding distribution of CDPs from South Georgia (blue, adapted from [24]) and from two colonies in North Island New Zealand (red; adapted from [40]); the blue and red arrows show the colonies where the species was studied in [24,40], respectively. The black lines represent the approximate location of the Subtropical Front (STF), Polar Front (PF) and Southern Antarctic Circumpolar Current Front (SACCF). The devices were deployed on breeding birds. For Kerguelen in 2015–2016, the GLS stopped recording soon after their deployments (two to four months), and therefore, the present figure describes a partial view of the CDP distribution for that year.

(the only year with data from the three sites), the three populations headed to different post-breeding migration areas within the Southern Ocean. Individuals from Kerguelen and Kanowna Island migrated south-southwest from their respective colonies, while individuals from Mana Island headed east-southeast. Individuals from Kerguelen migrated to more southerly latitudes (55–65° S) than those from both Kanowna Island (51–60° S) and Mana Island (51–59° S). All individuals started their post-breeding migration shortly after their last visit to the burrow (approx. 1–2 days after last burrow attendance). The timing of departure differed significantly among sites according to the differences in breeding phenology (table 1; Kruskal–Wallis test: $\chi^2 = 24.108$, $p < 0.001$). Thus, individuals from Kerguelen departed 107 and 80 days after birds from Kanowna Island and Mana Island, respectively.

The duration of the post-breeding migration was on average 120 and 92 days longer for birds from Kerguelen than from Kanowna and Mana Islands (table 1; Welch's ANOVA: $F_{10.976} = 94.321$, $p < 0.001$). It was also significantly different between the latter two populations (paired *t*-test: $t_{6.993} = -5.781$, $p < 0.001$). Most individuals were performing their post-breeding migration between March and September at Kerguelen, between November and February at Kanowna Island and between December and April at Mana Island (table 1 and figure 3). The total distance travelled during this period was the greatest for individuals from Kerguelen and the shortest for those from Kanowna Island (table 1). By contrast, maximum range from the colony was not related to either the total distance travelled or the duration of the post-breeding migration, with birds from Mana Island going the farthest from their colony and the birds from Kerguelen the nearest (table 1; Mann–Whitney *U*-test: for Mana Island–Kanowna Island $U = 9$, $p = 0.905$; for Kanowna Island–Kerguelen $U = 32$, $p = 0.018$).

Overall, the three populations of CDP displayed a similar activity pattern: they spent a relatively low proportion of time on the water at the start and end of the post-breeding migration (less than 70% of each day), and showed a peak of time spent on the water (greater than 90% of each day) 34–41 days after departure (figure 4 and table 1). The timing of this peak differed among populations (ANOVA: $F_{5.426} = 19.855$, $p = 0.002$), but the duration between departure and the maximum time spent on the water did not (ANOVA: $F_{5.007} = 0.435$, $p = 0.782$). In addition, the date of maximum time spent on the water was highly correlated with the date of departure for migration (electronic supplementary material, figure S1; Spearman's correlation test: $S = 189.510$, $p < 0.001$, $\rho = 0.927$).

## 3.2. Stable isotope values

Stable isotope values of body feathers (four per individual) varied substantially, from −26.0 to −15.9 ‰ for $\delta^{13}$C and from 3.9 to 15.2 ‰ for $\delta^{15}$N. Within each population, there were similar patterns, with feathers splitting into two groups (figure 5). The majority of feathers had low $\delta^{13}$C (from −26.1 to −19.5 ‰) and $\delta^{15}$N (from 3.8 to 9.9 ‰) values (group 1), while a smaller group of body feathers exhibited markedly higher isotopic values (from −19.5 to −15.9 ‰ for $\delta^{13}$C and from 9.9 to 15.1 ‰ for $\delta^{15}$N; group 2).

**Table 1.** Timing and duration of the post-breeding migration, and mean moult date of CDPs (*P. urinatrix*) from Kerguelen, Kanowna and Mana Islands (mean ± s.e.). (The mean moult date of flight feathers was identified as the peak of time spent on the water (greater than 90% time per day sitting on water). Birds were equipped with miniaturized saltwater immersion geolocators (see Methods).)

| | Kerguelen | | Kanowna | | Mana |
|---|---|---|---|---|---|
| | 2015–2016 (*n* = 5) | 2017–2018 (*n* = 7) | 2017–2018 (*n* = 5) | 2018–2019 (*n* = 8) | 2017–2018 (*n* = 4) |
| last burrow occupancy | 29 Feb ± 17 days | 26 Feb ± 5 days | 12 Nov ± 1 day | 16 Dec ± 3 days | 08 Dec ± 2 days |
| departure date | 2 Mar ± 16 days | 27 Feb ± 4 days | 12 Nov ± 1 day | 17 Dec ± 3 days | 9 Dec ± 2 days |
| outward travel duration (days) | 11.0 ± 2.3 | 15.3 ± 1.6 | 13.8 ± 2.1 | 12.4 ± 1.4 | 7.8 ± 0.8 |
| post-breeding migration duration (days) | — | 211 ± 6 | 92 ± 4 | 127 ± 8 | 119 ± 3 |
| inward travel duration (days) | — | 7.1 ± 1.6 | 9.6 ± 0.9 | 13.1 ± 3.6 | 9.5 ± 0.9 |
| return date | — | 23 Sep ± 3 days | 8 Feb ± 5 days | 12 Apr ± 6 days | 4 Apr ± 6 days |
| first burrow occupancy | — | 25 Sep ± 4 days | 11 Feb ± 5 days | 22 Apr ± 10 days | 6 Apr ± 5 days |
| maximum range (km) | — | 2514 ± 103 | 3153 ± 267 | 2769 ± 119 | 3599 ± 407 |
| total distance (km) | — | 44 232 ± 2218 | 31 603 ± 2561 | 33 309 ± 3550 | 33 514 ± 1033 |
| mean wing moult date | — | 10 Apr ± 5 days | 16 Dec ± 3 days | 23 Jan ± 3 days | 14 Jan ± 2 days |
| time between departure and mean wing moult (days) | — | 41 ± 5 | 34 ± 3 | 37 ± 2 | 36 ± 4 |

Both $\delta^{13}C$ and $\delta^{15}N$ mean values differed significantly among populations (Friedman test: $\chi^2 = 101$ for $\delta^{13}C$ and 52.2 for $\delta^{15}N$, both $p < 0.001$), with individuals from Kerguelen exhibiting the lowest values of $\delta^{13}C$ in group 1 but the highest in group 2. The $\delta^{15}N$ values for the individuals from Kanowna Island in group 1 had a larger range than in the other populations (figure 5), with extremely low $\delta^{15}N$ values in six body feathers (3.9 ‰ < $\delta^{15}N$ < 5.7 ‰) from four different individuals (two females and two males). The proportion of body feathers in group 2 varied among populations (3.4% for Kerguelen, 16.0% for Kanowna Island and 10.8% for Mana Island) and within populations (electronic supplementary material, figure S2), with either individuals with body feathers only in group 1 or in both groups 1 and 2 but never with all the four feathers in group 2. Values of $\delta^{13}C$ in body feathers of tracked individuals were correlated with latitude of the kernel UD centroid, $\delta^{13}C$ values decreasing with higher latitudes (electronic supplementary material, figure S3; Spearman's correlation test: $S = 44630$, $p < 0.001$, $\rho = 0.649$).

## 3.3. Inter-annual variations

Multi-year data were collected for GLS tracking (2015–2016 and 2017–2018 for Kerguelen; 2017–2018 and 2018–2019 for Kanowna Island; table 1), and stable isotopes (2015–2016, 2016–2017 and 2017–2018 for Kerguelen; 2016–2017, 2017–2018 and 2018–2019 for Kanowna Island; table 2). Trip parameters differed substantially between years at Kanowna Island ($p < 0.01$ in all cases), but not at Kerguelen (table 1). Specifically, individuals from Kanowna Island departed on average 35 days later, stayed away 35 days longer and returned to the breeding region 70 days later in 2018–2019 than in

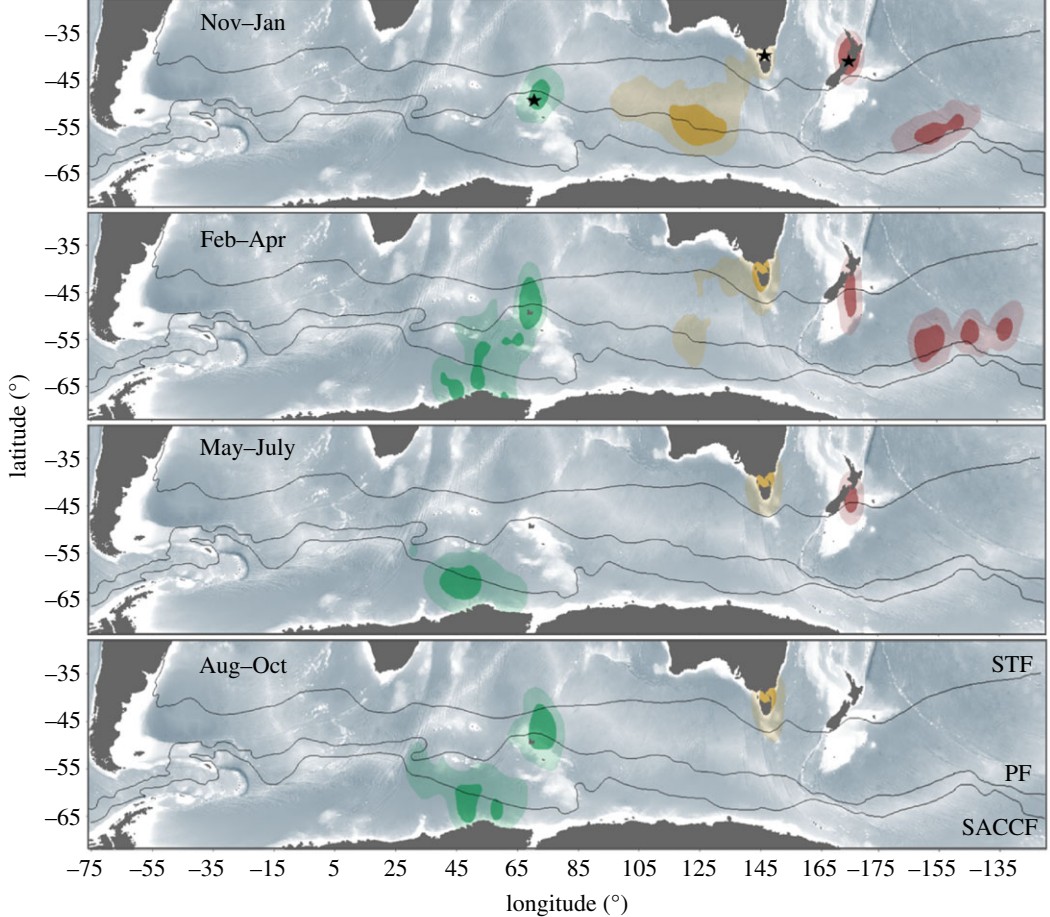

**Figure 3.** Year-round kernel density estimation derived from GLS locations of CDP (*P. urinatrix*) in 2017–2018 from Kerguelen (light green, $n = 7$), Kanowna Island (orange, $n = 5$) and Mana Island (red, $n = 4$). Solid colour areas show the 50% (core area) and faint colour the 95% (home range) of the kernel UD. The devices were deployed on breeding birds. Aug–Oct data were not available for Mana Island, as the GLS loggers were recovered in June 2018. See figure 1 for more details on the species phenology. The black lines represent the approximate location of the Subtropical Front (STF), Polar Front (PF) and Southern Antarctic Circumpolar Current Front (SACCF).

2017–2018. However, the maximum range and the total distance travelled were similar between the 2 years ($p > 0.2$ in both cases), as were post-breeding destinations (figure 2).

For both regions, feather $\delta^{13}$C values were generally consistent among years, except for Kerguelen between 2016–2017 and 2017–2018 (table 2; electronic supplementary material, figure S4; Mann–Whitney $U$-test: $U = 754.5$, $p = 0.002$), while $\delta^{15}$N values differed greatly among years for Kerguelen (Friedman test: $\chi^2 = 10.806$, $p = 0.004$) and Kanowna (Friedman test: $\chi^2 = 25.220$, $p < 0.001$), with for the latest a mean value of $\delta^{15}$N 1.8 ‰ lower in 2018–2019 than in 2017–2018 (table 2; Mann–Whitney $U$-test: $U = 582$, $p < 0.001$). Additionally, for Kanowna, the proportion of feathers in group 2 varied from 22.9% in 2016–2017 and 20.0% in 2017–2018 to 3.1% in 2018–2019 (electronic supplementary material, figure S5). Nonetheless, even when considering only the feathers from group 1, $\delta^{15}$N values were 1.4 ‰ lower in 2018–2019 than in 2017–2018 (Mann–Whitney $U$-test: $U = 457.5$, $p < 0.001$).

## 4. Discussion

By combining GLS tracking with stable isotope analysis of body feathers, the present study described the post-breeding movements and timing of moult of CDP from three populations breeding across two oceanic basins in the Southern Hemisphere. Despite the high wing loading of CDPs [51] and their high daily energy expenditure [64,65], all the individuals in the present study travelled long distances (20 000–55 000 km) during the post-breeding migration. Our results also show strong inter-population

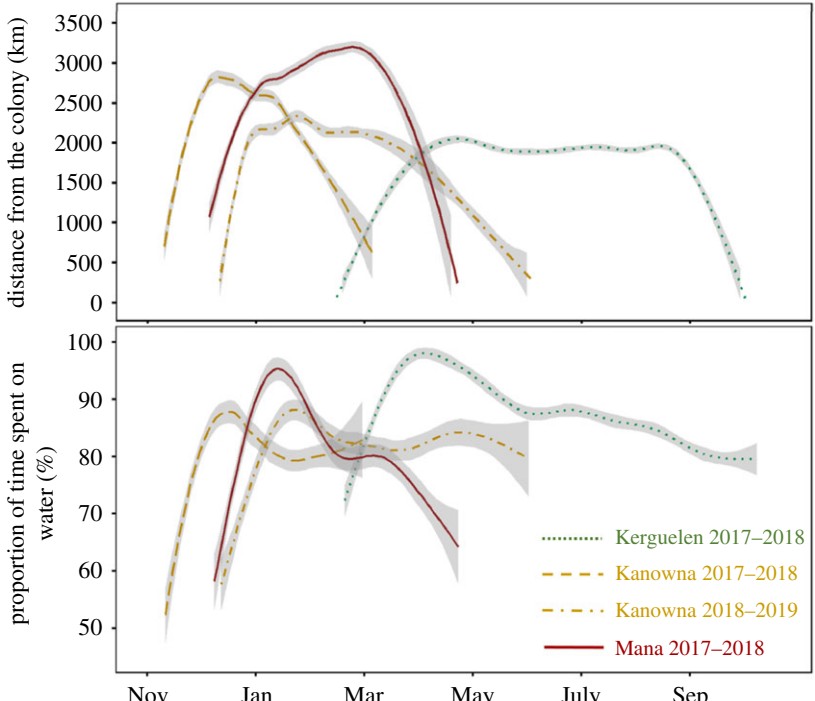

**Figure 4.** Distance from the colony (upper panel) and proportion of time spent on the water per day (lower panel) during the post-breeding migration of CDPs (*P. urinatrix*) from Kerguelen (green dotted line, 2017–2018, *n* = 7), Kanowna (orange long-dashed line, 2017–2018, *n* = 5; and orange dot-dashed line, 2018–2019, *n* = 8) and Mana Islands (red solid line, 2017–2018, *n* = 4). Data were fitted with a generalized additive mixed models. The shaded areas along the curves represent the 95% confidence interval.

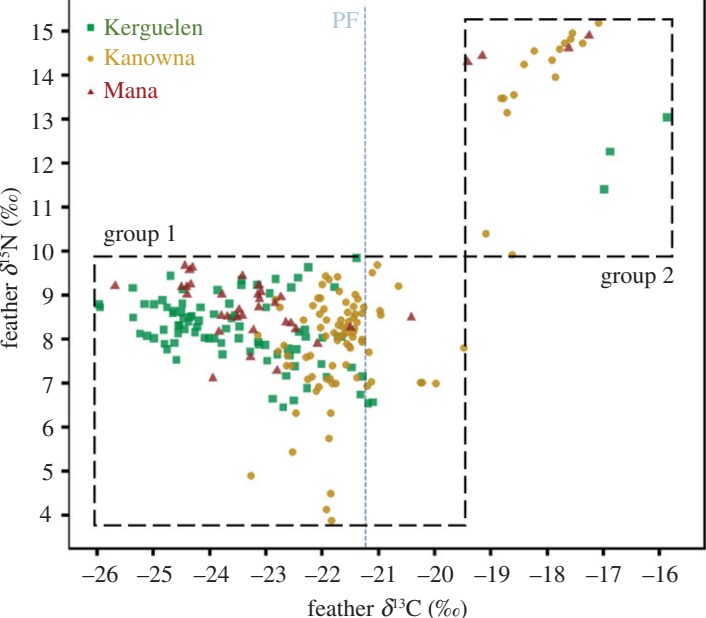

**Figure 5.** Body feather $\delta^{13}$C and $\delta^{15}$N values of CDPs (*P. urinatrix*) from Kerguelen (green squares, *n* = 23), Kanowna (orange dots, *n* = 25) and Mana Islands (red triangles, *n* = 10). Points represent isotopic values of individuals feathers (four body feathers per individual). Isotopic ratios in group 1 correspond to Antarctic/subantarctic waters and group 2 to subtropical/neritic waters [61]. The blue dotted line corresponds to the feather $\delta^{13}$C estimation of the Polar Front (PF) [61].

differences in the phenology, distribution and activity of CDP during the non-breeding period. This highlights the importance of using a multi-population approach in ecology in order to fully understand the mechanisms of species adaptation to local environmental variations.

**Table 2.** Body feather $\delta^{13}$C and $\delta^{15}$N values (‰) of CDPs (P. urinatrix) from Kerguelen, Kanowna and Mana Islands. (Values are means (±s.d.), and n and N refer to the number of feathers and of individuals (four feathers per individual), respectively.)

| | | Kerguelen | Kanowna | Mana |
|---|---|---|---|---|
| feather $\delta^{13}$C (‰) | 2015–2016 | −23.3 ± 1.6 | — | — |
| | | (n = 24; N = 6) | | |
| | 2016–2017 | −22.8 ± 1.9 | −20.9 ± 1.8 | — |
| | | (n = 40; N = 10) | (n = 48; N = 12) | |
| | 2017–2018 | −24.1 ± 1.2 | −21.1 ± 1.3 | −22.8 ± 1.8 |
| | | (n = 28; N = 7) | (n = 20; N = 5) | (n = 40; N = 10) |
| | 2018–2019 | — | −21.5 ± 0.8 | — |
| | | | (n = 32; N = 8) | |
| feather $\delta^{15}$N (‰) | 2015–2016 | 8.7 ± 0.9 | — | — |
| | | (n = 24; N = 6) | | |
| | 2016–2017 | 8.4 ± 1.2 | 9.3 ± 3.1 | — |
| | | (n = 40; N = 10) | (n = 48; N = 12) | |
| | 2017–2018 | 7.9 ± 0.7 | 9.4 ± 1.5 | 9.3 ± 1.9 |
| | | (n = 28; N = 7) | (n = 20; N = 5) | (n = 40; N = 10) |
| | 2018–2019 | — | 7.6 ± 1.4 | — |
| | | | (n = 32; N = 8) | |

## 4.1. Spatial distribution

CDPs from the three study regions all migrated to oceanic frontal systems of the Southern Ocean. However, their post-breeding destinations were markedly different among the three focal populations, and differed also from those of birds from South Georgia, which remained in the southern Atlantic Ocean (figure 2; [25]). Birds from Mana Island in New Zealand migrated southeast in accordance with previous findings from two colonies on the east and west coasts of North Island (figure 2; [40]). This suggests behavioural homogeneity within the New Zealand population.

Given their high wing loading and very limited soaring capabilities [66], the flight energy expenditure of CDP is relatively high [64]. Therefore, the different migratory destinations and path headings among the populations could reflect individuals exploiting the nearest productive habitats during the non-breeding period [25,40]. Indeed, CDP from South Georgia, southeastern Australia and New Zealand all travel to the highly productive Polar Front region [25,40,67]. The observation that individuals from New Zealand did not travel directly towards the Polar Front could reflect longitudinal differences in the productivity of this oceanographic feature [67]. Such regional variations have also been shown to play an important role in the distribution of grey petrels (Procellaria cinerea) and sooty shearwaters (Ardenna grisea) breeding in New Zealand [68,69].

Alternatively, the inter-population differences in post-breeding migratory destinations of CDP could reflect competition avoidance. Spatial segregation is often considered to be a mechanism to reduce competition between species and/or populations [17]. The Polar Front is known to be exploited by a wide range of other seabirds in the region [70–72], including the closely related South Georgian diving petrel (Pelecanoides georgicus; A. Fromant et al. 2018, unpublished data). Consequently, the observation that post-breeding CDP from Kerguelen do not target the closest Polar Front area, but head farther south, could be owing to intra- and interspecific competition occurring in the densely populated southern Indian Ocean [37].

## 4.2. Timing of migration and moult

Breeding phenology in seabirds has been shown to follow a gradient, with nesting periods being delayed with increasing latitude [73]. In the present study, there was a three to four months difference between the timing departure in migration of individuals from Kanowna Island and Kerguelen. However, only a 20

day shift in breeding phenology should be expected owing to the 10° latitudinal difference between the two sites (i.e. a 2 day delay per degree of latitude, [73,74]). Interestingly, within the Kerguelen Archipelago, CDP nesting and feeding in the Golfe du Morbihan (a closed coastal habitat) have been shown to breed one month later than the individuals nesting on offshore sites [37]. Additionally, in the present study, the breeding phenology of CDP from Kanowna Island varied substantially among years. These suggest that local oceanographic conditions of foraging areas during the breeding season play a key role in determining the phenology of CDP.

In the present study, a substantial increase in the proportion of time spent on water 30–40 days after migration departure was found in all sites and years. Such a pronounced pattern of activity has been associated with the renewal of flight feathers in several small procellariiform species [29,50]. Migratory diving birds, such as some species of alcids and diving petrels, are known to undergo a rapid wing moult in early post-breeding period, temporary affecting their flight ability [46]. The physiological cost of a quasi-flightless period seems to be mediated by their diving and feeding capacities that remain efficient during wing moult [75]. Although this result could simply indicate a period of intensive feeding, the strong correlation between the peak timing and the migration departure suggests that this event is strongly influenced by the breeding phenology. The onset of flight feather renewal in other Procellariiformes has similarly been related to the end of the breeding season [76].

The important intra-individual variance in isotopic values of body feathers indicates they grew in different locations and, therefore, at different periods from the end of the breeding season to throughout the non-breeding period [50,52,77]. The results indicate most body feathers moulted in Antarctic/subantarctic waters but a small proportion moulted in subtropical or coastal areas [21]. However, the proportion of body feathers with Antarctic $\delta^{13}$C values differed among the three study populations which could be related to the timing and duration of migration. Indeed, at Kanowna Island, inter-annual differences in isotopic values were associated with unprecedented changes in breeding phenology between consecutive seasons for this species (electronic supplementary material, figure S5; [78]). These unexpected results indicate that the moult of body feathers was delayed according to the change in breeding phenology. This reinforces the findings showing that the timing of breeding is an important factor in determining the moult schedule of body feathers [79]. Additionally, a longer post-breeding migration duration lowered the number of isotopic values corresponding to subtropical/neritic waters. This suggests that most of the body feathers were moulted within the first four months after the end of the breeding season.

## 4.3. Duration of the post-breeding migration

In addition to inter-population differences in the timing and destinations of post-breeding migration, the present study revealed important differences in its duration, with individuals at Kerguelen remaining away from the colony for four and three months longer than those at Kanowna Island and Mana Island. These results are consistent with previous studies indicating that adults are absent from the colony for six to eight months at South Georgia [80], Crozet [36] and Kerguelen Islands [81], while on New Zealand's North Island, birds are away for half this duration [40]. The CDP from southeastern Australia and New Zealand, then, come back and stay in the vicinity of their respective breeding colony five to six months before initiating the next breeding cycle.

For both southeastern Australia and New Zealand populations, a decrease in marine productivity in the area where they migrate in post-breeding [67] and/or an increase of food availability near their breeding grounds at the end of summer (February–March; [82]) could induce an early return from their migration area. Such a pattern has been observed in thin-billed prions (*Pachyptila belcheri*) from the Falkland/Malvinas Islands [14] that, in contrast with conspecifics from Kerguelen, return earlier to the breeding grounds from substantially shorter migration periods to distant open waters in order to exploit an extensive local shelf area.

Conversely, this difference could be linked to the predictability of food resources in the vicinity of the breeding areas [83]. In southeastern Australia and New Zealand, the locally abundant coastal krill *Nyctiphanes australis* [82] is a key food source for CDP [34] (C. M. Miskelly 2019, personal observation). However, this euphausiid undergoes extreme inter-annual variations in biomass and distribution, considerably affecting the reproductive success of various fish [82] and seabird species [84,85]. Populations spending a long period near the colony before the reproductive season are more likely to follow local environmental cues to adapt their phenology [83], which may explain the significant delay in the breeding period observed between the years in the present study at Kanowna Island [78].

By contrast, CDP at Kerguelen and South Georgia, where marine productivity is relatively more predictable [32,86], may be able to return from migration closer to the mating period.

## 4.4. Implications for conservation

Six factors have been considered as potential barriers to gene flow between seabird populations: physical isolation; contrasted breeding phenology; strong philopatry; differences in ocean regime; and divergence in breeding and non-breeding distributions [11]. In addition to the obvious physical isolation between the main CDP populations and their small range in their foraging areas during the breeding period [87,88], the present study has provided evidence of segregation in the post-breeding migratory destinations of eastern CDP populations. Similarly, there are inter-population differences in diving behaviour [81,89,90], diet [32–34,91] and phenology [47,78,92,93], though these differences may be mainly related to the local oceanographic environment during the breeding season. Finally, because CDP are highly philopatric [94–96], the results of the present study provide further evidence that the species has a high potential for population differentiation [4]. Indeed, Cristofari *et al.* [13] recently revealed significant isolation and low inter-population flux in Peruvian diving petrel (*Pelecanoides garnotii*) colonies at very short distances (350 km).

Conservation biology often considers species as single taxonomical, biological and ecological entities, and not considering the presence of subspecies may mislead in determining the species' status and, thus, appropriate conservation strategies [15]. It is, therefore, of importance to revisit the genetic and taxonomic relationships among the different populations of CDP, as recently done for the South Georgian diving petrel [12,97] and the Peruvian diving petrel [13]. Meanwhile, caveats remain for the spatial distribution of juveniles and failed-breeders. Natal dispersal is a major factor influencing population structure and metapopulation dynamics [98], and despite the CDP being highly philopatric [94–96], knowledge of first-year individuals could have important implications for conservation.

In summary, the present study has provided clear evidence that CDP travel to Antarctic or subantarctic productive waters during the post-breeding migration. However, major temporal, spatial and trophic differences throughout the species' distribution highlight the relevance of multi-population studies [9]. Further information is required in order to better understand the genetic and taxonomic structure [17], and the degree of isolation, of subpopulations/subspecies [99]. It is notably vital for small remote populations such as those from Gough Island, Marion Island, Macquarie Island and Campbell Island that are or have been exposed to the threat of invasive rodents [100–102]. The approach used in the present study, investigating the phenology and distribution outside the breeding season to evaluate the population diversity, could be expanded to a wide range of seabird species, especially the under-studied small-sized Procellariiformes [17]. This could improve the knowledge of evolutionary divergence among populations [103] and assist in the challenging conservation of migratory species [104].

Ethics. All animal handling and instrumentation was approved by the Ethical Committee of the Terres Australes et Antarctiques Francaises, the New Zealand Department of Conservation, Deakin Animal Ethics (approval B16-2017) and DELWP Wildlife Research (permit 100084552).

Data accessibility. Our data are available within the Dryad Digital Repository: https://doi:10.5061/dryad.fbg79cnrz [105].

Authors' contributions. A.F. carried out the stable isotopes and statistical analysis, participated in the design of the study, collected field data and drafted the manuscript; K.D. participated in the data analysis, collected field data, helped draft the manuscript and critically revised the manuscript; A.C. participated in the stable isotopes analysis, collected field data and critically revised the manuscript; Y.H.E. participated in collecting field data, and critically revised the manuscript; P.B. and Y.C. helped in the data analysis, helped draft the manuscript and critically revised the manuscript; J.P.Y.A., C.-A.B. and C.M.M. conceived and coordinated the study, collected field data and critically revised the manuscript.

Competing interests. We declare we have no competing interests.

Funding. This study was supported logistically and financially by Sea World Research and Rescue Foundation Inc., Birdlife Australia, Museum of New Zealand Te Papa Tongarewa, the Institut Polaire Français Paul Emile Victor (Programme Oiseaux Plongeurs no. 394, C.-A.B.) and the Terres Australes et Antarctiques Françaises.

Acknowledgements. The authors thank G. Guillou and M. Brault-Favrou from the Plateforme Analyses Isotopiques of the LIENSs laboratory for running stable isotope analysis, C. Ribout for seabird molecular sexing, the fieldworkers (S. Peroteau, R. Debenest, J. Fleureau, U. Courcoux-Caro, T. Getti), the numerous field volunteers (e.g. A. Lec'hvien and B. Gardner) and P. Tixier for his help with R code maps. Thanks to the CPER (Contrat de Projet Etat-Région) and the FEDER (Fonds Européen de Développement Régional) for funding the IRMS of LIENSs laboratory. The IUF (Institut Universitaire de France) is acknowledged for its support to P.B. as a Senior Member.

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
