## [Reviewer comments · Royal Society Open Science]

Review History

RSOS-200670.R0 (Original submission)

Review form: Reviewer 1

Is the manuscript scientifically sound in its present form?

Yes

Are the interpretations and conclusions justified by the results?

Yes

Is the language acceptable?

Yes

Do you have any ethical concerns with this paper?

No

Have you any concerns about statistical analyses in this paper?

No

Recommendation?

Accept with minor revision (please list in comments)

Comments to the Author(s)

In this paper, the authors describe the spatial movements during the migration and the moulting patterns of a small seabird (common diving petrel) from three different breeding colonies covering a vast area of its breeding distribution. For this, they combine tracking devices (GLS) and stable isotopic data. Although the paper has been well written and, in my opinion, all the results are accurately described and discussed, I suggest reducing the length of the discussion section, trying to be more synthetic.

Review form: Reviewer 2**Is the manuscript scientifically sound in its present form?**

Yes

Are the interpretations and conclusions justified by the results?

No

Is the language acceptable?

No

Do you have any ethical concerns with this paper?

No

Have you any concerns about statistical analyses in this paper?

Yes

Recommendation?

Major revision is needed (please make suggestions in comments)

Comments to the Author(s)

See attached comments (Appendix A).

Decision letter (RSOS-200670.R0)

Dear Mr Fromant,

The editors assigned to your paper ("Temporal and spatial differences in the post-breeding behaviour of a ubiquitous Southern Hemisphere seabird, the common diving petrel") have now received comments from reviewers. We would like you to revise your paper in accordance with the referee and Associate Editor suggestions which can be found below (not including confidential reports to the Editor). Please note this decision does not guarantee eventual acceptance.

Please submit a copy of your revised paper before 30-Jul-2020. Please note that the revision deadline will expire at 00.00am on this date. If we do not hear from you within this time then it

will be assumed that the paper has been withdrawn. In exceptional circumstances, extensions may be possible if agreed with the Editorial Office in advance. We do not allow multiple rounds of revision so we urge you to make every effort to fully address all of the comments at this stage. If deemed necessary by the Editors, your manuscript will be sent back to one or more of the original reviewers for assessment. If the original reviewers are not available, we may invite new reviewers.

- Data accessibility

If you wish to submit your supporting data or code to Dryad (<http://datadryad.org/>), or modify your current submission to dryad, please use the following link:
<http://datadryad.org/submit?journalID=RSOS&manu=RSOS-200670>

- Competing interests

- Authors' contributions

- Acknowledgements

- Funding statement

Kind regards,
Lianne Parkhouse
Editorial Coordinator
Royal Society Open Science
openscience@royalsociety.org

on behalf of Dr Denise Greig (Associate Editor) and Pete Smith (Subject Editor)
openscience@royalsociety.org

Associate Editor's comments (Dr Denise Greig):

Associate Editor: 1

Comments to the Author:

The authors use light sensing dataloggers and stable isotope ratios to evaluate movements and make inferences about behavior for common diving petrels from three geographically distinct breeding locations in the Southern Ocean. I agree with the authors that it is important to consider full range of movements, not just breeding sites, when assessing conservation concerns as well as resilience to climate change. These data documenting petrel movement when away from the breeding site are very interesting.

The spatial analyses from the GLS tags are well done and the figures are great, but how these data work in tandem with the stable isotope data is a little confusing.

I am also unclear from your descriptions whether the wet-dry sensors are really telling you when the birds moult.

Page 10 – the first paragraph starts by saying you used the wet-dry sensor to determine when moult occurred, but do not list this at the end of the paragraph where you listed the parameters you determined using the wet dry sensor data. Aside from that, maybe it would be more accurate to report the timing when petrels spent greater than 90% of their time on the water (in the results and in Table 1). Then say that you used this as a proxy for the moult. Is it possible though that it could be something else, like a prey aggregation? It is hard to assess without seeing the tracking data or having some other information about these locations where they spent time on the water. Is there any other way to know that moult occurs during the inter breeding period? Does it occur every year? etc. I think that information would be useful when you talk about the sampling of the feathers for stable isotopes on feathers that you assume were grown during the inter breeding period.

Also, in the discussion (page 18, lines 347-352), you say the reports on timing of moult vary in the literature, but it is unclear how much. is sometimes observed on the breeding colonies? Or are the differences more like beginning versus end of the inter breeding period?

I could not tell what information you gained from the stable isotope ratios that you don't already get from the GLS tags: in the abstract you suggest trophic differences, but in the methods it sounds like you are getting additional location information. It would be interesting to know if there were instances where the broad locations from the SI data did not match the GLS data. And if you are interested in trophic differences do you mean between breeding and non-breeding petrels? Or the petrels at three different locations?

Please revise instances where you use the word "respectively":

For example, in the Figure 5 caption, you could rewrite as "Isotopic ratios from Group 1 feathers correspond to Antarctic/subantarctic waters and Group 2 to subtropical/neritic waters.

Please re-write Page 18, lines 366-369 without using "respectively" - the sentence is very convoluted as it is.

Page 19, line 376. Simply delete the word "respectively"

Reviewers' Comments to Author:

Reviewer: 1

Comments to the Author(s)

In this paper, the authors describe the spatial movements during the migration and the moulting patterns of a small seabird (common diving petrel) from three different breeding colonies covering a vast area of its breeding distribution. For this, they combine tracking devices (GLS) and stable isotopic data. Although the paper has been well written and, in my opinion, all the results are accurately described and discussed, I suggest reducing the length of the discussion section, trying to be more synthetic.

Reviewer: 2

Comments to the Author(s)

See attached comments.

Author's Response to Decision Letter for (RSOS-200670.R0)

See Appendix B.

RSOS-200670.R1 (Revision)

Review form: Reviewer 1

Is the manuscript scientifically sound in its present form?

Yes

Are the interpretations and conclusions justified by the results?

Yes

Is the language acceptable?

Yes

Do you have any ethical concerns with this paper?

Yes

Have you any concerns about statistical analyses in this paper?

No

Recommendation?

Accept as is

Comments to the Author(s)

After this review, the authors answered most of the suggestions provided by the reviewers and now the ms is more clear and interesting.

Decision letter (RSOS-200670.R1)

Dear Mr Fromant

On behalf of the Editors, we are pleased to inform you that your Manuscript RSOS-200670.R1 "Temporal and spatial differences in the post-breeding behaviour of a ubiquitous Southern Hemisphere seabird, the common diving petrel" has been accepted for publication in Royal Society Open Science subject to minor revision in accordance with the referees' reports. Please find the referees' comments along with any feedback from the Editors below my signature.

Please submit your revised manuscript and required files (see below) no later than 7 days from today's (ie 22-Oct-2020) date. Note: the ScholarOne system will 'lock' if submission of the revision is attempted 7 or more days after the deadline. If you do not think you will be able to meet this deadline please contact the editorial office immediately.

on behalf of Dr Denise Greig (Associate Editor) and Pete Smith (Subject Editor)
openscience@royalsociety.org

Associate Editor Comments to Author (Dr Denise Greig):

Thank you for addressing previous reviewer comments. I just have a few minor edits in track changes in the attached file.

Reviewer comments to Author:

Reviewer: 1

Comments to the Author(s)

After this review, the authors answered most of the suggestions provided by the reviewers and now the ms is more clear and interesting.

===PREPARING YOUR MANUSCRIPT===

- one version identifying all the changes that have been made (for instance, in coloured highlight, in bold text, or tracked changes);
- a 'clean' version of the new manuscript that incorporates the changes made, but does not highlight them.

This version will be used for typesetting.

===PREPARING YOUR REVISION IN SCHOLARONE===

Author's Response to Decision Letter for (RSOS-200670.R1)

See Appendix C.

Decision letter (RSOS-200670.R2)

Dear Mr Fromant,

It is a pleasure to accept your manuscript entitled "Temporal and spatial differences in the post-breeding behaviour of a ubiquitous Southern Hemisphere seabird, the common diving petrel" in its current form for publication in Royal Society Open Science.

on behalf of Dr Denise Greig (Associate Editor) and Pete Smith (Subject Editor)
openscience@royalsociety.org

Appendix A

Reviewer's comments – Fromant et al Common Diving-petrel

Line 3. I question the proposition that most seabirds “migrate” – the paradigm of migration is fixed route – most seabirds don't undertake fixed route migrations – rather there is a post-breeding dispersal of all age classes. Arctic Terns are defensible as migratory species – I would like to see evidence of it in CDP – rather than Kernel Analyses. Too many studies in last decade or so confound migration with dispersion – the two are very different behaviours.

Line 8. Re demographic processes/parameters – see Price et al 2020 MEPS re 70-year data set for Short-tailed Shearwaters and link between demographics and oceanographic processes.

Line 16 and throughout. I must confess I dislike the term “inter-breeding period” – it's not a term I have seen used – the normal phrase is either ‘post-breeding’ or ‘non-breeding’. Inter-breeding seems contrived and awkward compared to the extant literature. I note at Lines 51 and 56, the authors use ‘post-breeding’ – which is the convention.

Lines 19-24. Authors haven't convinced me that the IBP (inter-breeding period) is more ‘critical’ than the breeding period. I don't have a problem with the rest of the argument lines 18-19 re population-specific ecologies, but why is IBP ‘critical’?

Line 40. Suggest authors cite BirdLife International's seabird tracking database rather than 2 relatively constrained studies.

Line 47. Replace ‘between’ with ‘among’; between refers to 2 options, among is for 3 or more. Ditto Lines 62 and 182 and 254 and 260 and 271 and etc.

Line 53. There are other Indian Ocean populations of CDP beyond Kerguelen and Iles Crozet – suggest authors reference studies from these breeding sites, some of which meet the criteria of ‘large’.

Line 67. You are more likely to see ‘divergence’ in three widely-spaced study populations than if the study was conducted on 3 closely-spaced populations. This also was raised in the Abstract (line #22). I can imagine very different results if you had looked at Marion/Crozet/Kerguelen/Heard populations of CDP.

Line 79. Replace ‘further’ with ‘farther’ (as you are talking about distance). Ditto line 210. Line 232 is ‘farthest’.

Lines 81 – 88 – these definitions do not concur with broader behavioural ecological literature. Dingle & Drake 2007 is not the sole citation for migration; see also my previous comments re use of IBP. I would encourage authors to adopt terminology that is widely used.

Line 93. A 1.5g GLS unit is more than 1% of a 110g individual. I would like to see data on body masses of individuals used presented in the Supplementary Materials at the very least.

Line 107 – there are molt details for Heard I in the primary literature – rather than HANZAB.

Line 111. I don't necessarily agree that more time on water reflects molt in CDP (see below).

Line 134. Unrealistic distance not speed is >1500km/day.

Lines 135-136. Authors describe beginning of outward and end of inward – what about the end of outward or the start of inward trips? If these birds are “migrating” (which they are not) these dates could be defined. However, as the CDP are dispersing post-breeding rather than migrating, these dates are impossible to define and the kernel analyses reinforces my concerns re confounding of dispersal and migration.

Line 150. There is a circularity in the assumption that more time spent sitting on water corresponds with molt, and that molt decreases flying efficiency and therefore the CDP will spend more time sitting on water. The 2 cited studies are for much larger flying birds and it is questionable that the results for albatrosses are true for CDP. CDP spend plenty of time on the water close to colonies and close to prey patches such as slicks and fronts, so I suspect there is greater behavioral variability in CDP to explain on water as solely limited to molt periods.

(and continuing at Lines 236-244. The periods at sea and on water correspond with the Southern Ocean winter – poor weather, especially close to SACCF). The birds may well be sheltering on the surface that trying to fly around in turbulent air, and they are likely to be feeding to deposit body reserves for the coming breeding season. I don't believe they are constrained in flight abilities due to molt. You could speculate but resting/feeding behaviors are equally compelling hypotheses)

Line 167. Replace 'prior to' with 'before'. One has prior knowledge, one does nothing 'prior to' anything. Ditto line 381.

Line 185. Please justify the selection of Gaussian models, and whether the associated assumptions were met.

Lines 188 – 192. When were parametric and non-parametric analyses undertaken? Would be useful to describe explicitly for readers.

Line 210. SACCF – missing 'Current' in text.

Lines 213 – 222. The use of atmospheric fronts by burrowing petrels is well known – the birds wait until after the passage of a cold front over the breeding island, then depart on foraging trips or longer – taking advantage of a SW air flow after the passage of the front – see the publications from Marion I. See also the work on short-tailed shearwaters (Raymond et al, Woehler et al, Schaeffer et al) as examples. I suspect (Line 218) colony departures were linked with breeding phenology AND local weather systems.

The most interesting aspect of Figure 2 is why do Mana Island CDP fly SE rather than SW into prevailing winds? Either wind flow regionally is different and/or the area to where the CDP disperse is more productive than other areas within the same dispersion distance.

Line 256. Please remove 'interestingly' as subjective. Ditto Line 294 'outstanding'. Ditto line 354 'profound'. Line 374 'stark'.

Lines 267 – 287. The question of intra- versus inter-annual variabilities needs further discussion. If intra-annual variability is greater than inter-annual, it will be hard to make generalisations about the one year when data are available from all three sites. Are there independent data on productivity that could be used to examine the annual isotope signatures? How great are the associated latitudinal differences associated with the stated differences in $\Delta^{15}N$?

Line 292 (and 309). Why is wing loading "important"?

Line 301. The CDP dispersed TO or AROUND oceanic frontal systems, not in.

Lines 301 – 307. The inter-island dispersions reflect post-breeding dispersion to nearest high(er) productivity areas in adjacent areas of the Southern Ocean where the CDP could undertake molt and deposit reserves for the next breeding season. Line 307 – "behavioural homogeneity" – the productive areas used are simply within foraging/dispersal capacities of CDP colonies in New Zealand.

Lines 320 – 330. Suggest delete and confine discussion to previous paragraph re oceanic productivity. The text is speculative with no supporting data.

Line 342. What about founder effects potentially generating population-specific phenologies?

Line 385. Are there any independent data to suggest a decrease in marine productivity?

Line 399. Nocturnal vocalisations at colonies are both territorial activity and also displaying/soliciting mates.

Lines 405 – 409 re *Nyctiphanes*. The inter-annual variability in *Nyctiphanes* abundance is driven by regional winds driven by ENSO events – see Mills et al ref #78. Kerguelen, Kanowna and South Georgia etc CDP populations all influenced by ENSO and/or ACW and/or other dipole oceanographic/atmospheric phenomena.

Lines 383 – 412 are weak and I suggest removing.

Line 419 – As I understand it, the study looked at breeding CDP? What about dispersal of non-breeding CDB between fledging and initial breeding effort? These birds will not be constrained to a colony and could potentially disperse (not migrate) farther.

Line 420. The study has not “provided clear evidence of segregation of . . . CDP populations”. The study has provided clear evidence of segregation of widely-separated CDP populations – which is not the same thing. Useful and interesting results would emerge from studies on closer-spaced populations – if they behave similarly (as you might expect), then that shows some homogeneity. If not, the story could explore competition etc.

Figure 3. Presumably these are for breeding adults, so the caption should state this explicitly. Why are year-round kernels smaller and substantially shifted compared to the winter kernels?

Supplementary Figures S1 – S5. Please state sample sizes n explicitly in all figure legends.

Appendix B

Response to the editor and reviewers

Please find below the detailed responses to the editor and reviewers (in green). All the comments (from the editor and the two reviewers) were addressed point by point.

In a first section we responded to the general comments that was raised by the editor or reviewers.

The second section addresses the comments point by point.

General response to the comments:

The editor and reviewer 2 raised concerns regarding whether the wet-dry sensors are really telling when the birds moult, and instead, could also reflect a resting/feeding behaviour related to prey aggregation and/or poor weather conditions.

Response: More information about the moult of flight feathers and body feathers were added in the Methods. Additionally, this question was further developed in the Discussion.

Nevertheless, we believe that our results validate the theory of “moult of flight feathers”:

- Feeding versus moulting. The most important thing is that the two activities are not exclusive (Bridge 2004). Birds can (and have to) moult wing feathers and feed at the same time. Moult is costly, and thus birds have to feed a lot during moult, except if they stored large energy reserves before moult (penguins, grebes), which is not the case for diving petrels. Alcids illustrate this issue well: they are diving seabirds with most of them going through a long period of flightlessness after leaving the colony. Feeding a lot is especially important with the males that rear their single chick whilst moulting flight feathers at sea.
- In Fig. 4 lower panel, the results highlight that a peak of time spent on the water when the birds arrive in the post-breeding migration area. If this peak was only related to feeding behaviour, the curve should reach a plateau without major variation between the outward and inward movements (similar to the curve of distance from the colony, Fig. 4 upper panel).
- Reviewer 2 suggested that, during Southern Ocean winter (poor weather conditions), the birds may well be sheltering on the surface rather than trying to fly around in turbulent air. Although it is an interesting theory, similarly to the previous argument, this does not explain the peak of time spent on the water. The timing of the peak and the strong correlation with the end of breeding season does not match with a stochastic event. Additionally, in the example of individuals from Kerguelen, the peak is in April-May (austral autumn), and not throughout the winter. For the 2 other populations, the peak is during the austral summer.

Reviewer 2 raised concerns regarding of a potential confounding of dispersal and migration.

Response: the different results gathered in the previous and the present studies suggest that adult CDP migrate (and not disperse) after the end of the breeding season, and this for several reasons:

- As highlighted in Fig. 4 (upper panel), the post-breeding “migration” is clearly delimited by rapid outward and inward movements, separated by a period of a relatively constant distance from the colony (2 000 to 3 000 km away from the colony).
Dispersal would imply a smoothed curve without clear outward and inward movements.
- The consistency in direction path and “migration” area within each population (and inter-annually) highlight a pattern that is hardly explainable with the dispersal theory. Post-breeding dispersal should reflect a greater inter-individual variation within each population. The New Zealand population illustrates this paradigm very well with individuals from 3 islands and 2 different years (present study and Rayner et al. 2017) that all went in the same direction and stopped in the same area.
- The hypothesis of a similar inter-individual dispersion taking advantage of the SW air flow (potentially explaining the movement SW of birds from Kerguelen and Kanowna Island) does not match with the direction taken by the birds from South-Georgia (East; Navarro et al. 2015) and New Zealand (South-West; present study and Rayner et al. 2017).

Associate Editor's comments (Dr Denise Greig):

Associate Editor: 1

Comments to the Author:

The authors use light sensing dataloggers and stable isotope ratios to evaluate movements and make inferences about behavior for common diving petrels from three geographically distinct breeding locations in the Southern Ocean. I agree with the authors that it is important to consider full range of movements, not just breeding sites, when assessing conservation concerns as well as resilience to climate change. These data documenting petrel movement when away from the breeding site are very interesting.

The spatial analyses from the GLS tags are well done and the figures are great, but how these data work in tandem with the stable isotope data is a little confusing.

I am also unclear from your descriptions whether the wet-dry sensors are really telling you when the birds moult.

Response: The Methods and Discussion were updated in order to provide more information and discussion about this point. Please see also the detailed response in the general comments section.

Page 10 – the first paragraph starts by saying you used the wet-dry sensor to determine when moult occurred, but do not list this at the end of the paragraph where you listed the parameters you determined using the wet dry sensor data. Aside from that, maybe it would be more accurate to report the timing when petrels spent greater than 90% of their time on the water (in the results and in Table 1). Then say that you used this as a proxy for the moult. Is it possible though that it could be something else, like a prey aggregation? It is hard to assess

without seeing the tracking data or having some other information about these locations where they spent time on the water.

Response: The paragraph page 10 was split in 2 different paragraphs in order to facilitate the reading. In the first paragraph, we describe the method to identify the moult of flight feathers using the wet/dry sensor only. The list of parameters at the end of the second paragraph are determined in order to describe and detail the migration characteristics (timing and distance). To do so, we used in combination the wet/dry sensor, temperature sensor, and location.

Additionally, more information about the moult of flight feathers and body feathers was added in the Methods. The text was updated throughout the manuscript in order to make a clear distinction between the results about the moult of flight feathers (using the wet/dry sensor) and the moult of body feathers (using stable isotope analysis).

Figure S2 was updated in order to better describe when and where the moult of flight feathers occurred.

The section *Timing of migration and moult* in the Discussion was restructured in order to discuss about the relation between time spent on the water and feeding behaviour.

Is there any other way to know that moult occurs during the inter breeding period? Does it occur every year? etc. I think that information would be useful when you talk about the sampling of the feathers for stable isotopes on feathers that you assume were grown during the inter breeding period.

Response: More details about the moult of Pelecanoidae species and CDP was added in the Methods section (paragraph line 115-122 for body feathers moult). Additionally, the paragraphs were restructured in order to clarify the difference between wing moult and body feathers moult.

Also, in the discussion (page 18, lines 347-352), you say the reports on timing of moult vary in the literature, but it is unclear how much. is sometimes observed on the breeding colonies? Or are the differences more like beginning versus end of the inter breeding period?

Response: More details about the moult of flight feathers and body feathers were added in the Methods section. As mentioned, despite the lack of information, the same pattern seems to be common to all diving petrel species, with a quick wing moult just after the breeding season. For the body feathers, the renewal seems to be spread over a longer duration throughout the non-breeding period (similarly to most of small procellariiform species).

In the present paper, our results confirmed these statements:

- The peak of time spent on the water (using the wet/dry sensor) just after the end of the breeding season suggests the moult of flight feathers
- The intra-individual variation in stable isotopic values confirms that the body feathers are renewed throughout the non-breeding season.

I could not tell what information you gained from the stable isotope ratios that you don't already get from the GLS tags: in the abstract you suggest trophic differences, but in the methods it sounds like you are getting additional location information. It would be interesting

to know if there were instances where the broad locations from the SI data did not match the GLS data. And if you are interested in trophic differences do you mean between breeding and non-breeding petrels? Or the petrels at three different locations?

Response: By combining GLS and stable isotope analysis, these results provide important information on the moult of body feathers. The 2 methods are complementary, highlighting that, for the 3 populations, most of the body feathers were synthesised during the migration period (see Fig. 5 and Fig. S2). The SIA results also indicate that some of the body feathers are moulted in the vicinity of the breeding site (validated by the GLS data), the proportion of these feathers moulted outside the migration probably being related to the total duration of the migration (see Discussion in section *Timing of migration and moult*). Additionally, the inter-annual variation of SIA in the body feathers of CDP from Kanowna Island might suggest that most of the body feathers are renewed within the first 4 months of the non-breeding period (see Fig. S5 and Discussion in section *Timing of migration and moult*).

The section *Stable isotope values* in the Results describes the positive correlation between the latitude of the kernel UD and the values of $\delta^{13}\text{C}$ in body feathers. This result is presented in Fig. S3 in the supplementary data.

The abstract was rephrased in order to better match with the main outcome of the stable isotope analysis (provide information on the moult pattern of body feathers in CDP).

Please revise instances where you use the word “respectively”:

Response: the use of the word “respectively” was restricted to the minimum, and sentences were rephrased where this word was removed.

For example, in the Figure 5 caption, you could rewrite as “Isotopic ratios from Group 1 feathers correspond to Antarctic/subantarctic waters and Group 2 to subtropical/neritic waters.

Response: the sentence was rephrased “Isotopic ratios in Group 1 correspond to Antarctic/subantarctic waters and Group 2 to sub-tropical/neritic waters”

Please re-write Page 18, lines 366-369 without using “respectively” – the sentence is very convoluted as it is.

Response: the sentence was rephrased.

Page 19, line 376. Simply delete the word “respectively”

Response: the word “respectively” was removed

Reviewer: 1

Comments to the Author(s)

In this paper, the authors describe the spatial movements during the migration and the moulting patterns of a small seabird (common diving petrel) from three different breeding colonies covering a vast area of its breeding distribution. For this, they combine tracking devices (GLS) and stable isotopic data. Although the paper has been well written and, in my

opinion, all the results are accurately described and discussed, I suggest reducing the length of the discussion section, trying to be more synthetic.

Response: The discussion was reduced and some sections were restructured.

Reviewer: 2

Comments to the Author(s)

Line 3. I question the proposition that most seabirds “migrate” – the paradigm of migration is fixed route – most seabirds don’t undertake fixed route migrations – rather there is a post-breeding dispersal of all age classes. Arctic Terns are defensible as migratory species – I would like to see evidence of it in CDP – rather than Kernel Analyses. Too many studies in last decade or so confound migration with dispersion – the two are very different behaviours.

Response: as presented in the general comments section we believe that “migration” is the appropriate term. We support this analysis with our results that match the ecological definition of migration (according the extant literature): the clear and rapid outward and inward movements (see Fig. 4, Fig. S2 and Table 1), the consistent migration area (see Fig. 2) and the direction of migration within each population.

Line 8. Re demographic processes/parameters – see Price et al 2020 MEPS re 70-year data set for Short-tailed Shearwaters and link between demographics and oceanographic processes.

Response: this reference was added in the text.

Line 16 and throughout. I must confess I dislike the term “inter-breeding period” – it’s not a term I have seen used – the normal phrase is either ‘post-breeding’ or ‘non-breeding’. Inter-breeding seems contrived and awkward compared to the extant literature. I note at Lines 51 and 56, the authors use ‘post-breeding’ – which is the convention.

Response: as suggested, the term “inter-breeding migration” was replaced by “post-breeding migration”, and “inter-breeding period” was replaced by “non-breeding period”. Both terms are described in more details in the Methods.

Lines 19-24. Authors haven’t convinced me that the IBP (inter-breeding period) is more ‘critical’ than the breeding period. I don’t have a problem with the rest of the argument lines 18-19 re population-specific ecologies, but why is IBP ‘critical’?

Response: the importance of inter-breeding period in the life cycle of seabirds was added in the first paragraph.

Line 40. Suggest authors cite BirdLife International’s seabird tracking database rather than 2 relatively constrained studies.

Response: the 2 references were replaced by BirdLife International (2020)

Line 47. Replace ‘between’ with ‘among’; between refers to 2 options, among is for 3 or more. Ditto Lines 62 and 182 and 254 and 260 and 271 and etc.

Response: the word “between” was replaced by “among” where it was necessary.

Line 53. There are other Indian Ocean populations of CDP beyond Kerguelen and Iles Crozet – suggest authors reference studies from these breeding sites, some of which meet the criteria of ‘large’.

Response: detailed references were added for Prince Edwards/Crozet/Kerguelen/Heard.

Line 67. You are more likely to see ‘divergence’ in three widely-spaced study populations than if the study was conducted on 3 closely-spaced populations. This also was raised in the Abstract (line #22). I can imagine very different results if you had looked at Marion/Crozet/Kerguelen/Heard populations of CDP.

Response: It is clear that the physical distance between 2 populations is an important factor leading to population divergence (this point is discussed in the section *Implications for conservation* in the Discussion). Nevertheless, the 3 populations studied in this paper are physically isolated from each other (5 600 km between Kerguelen/Heard and Bass Strait; 2 000 km between Bass Strait and Cook Strait), and because there is no other population/island in between to each other, the distance aspect is very important.

It could be possible that the results would be different when studying less distant populations (such as Marion/Crozet/Kerguelen/Heard), and it would be a very interesting project (even if the logistic would be a major issue), but this is not the aim of the present study.

Additionally, Cristofari et al. (2019) studied the population fragmentation of Peruvian diving-petrel and described a strong genetic isolation at a small spatial scale (few hundred km) compare to the distribution of CDP populations.

Rayner et al. 2011 also provided a good example of divergence of Cook’s petrel populations in New Zealand.

Line 79. Replace ‘further’ with ‘farther’ (as you are talking about distance). Ditto line 210. Line 232 is ‘farthest’.

Response: The words “further” and “furthest” were replaced accordingly.

Lines 81 – 88 – these definitions do not concur with broader behavioural ecological literature. Dingle & Drake 2007 is not the sole citation for migration; see also my previous comments re use of IBP. I would encourage authors to adopt terminology that is widely used.

Response: We agree that there is an extended literature about the behavioural ecological definition of migration. However, to our point of view, Dingle & Drake 2007 appears as a legitimate reference (widely recognised by the broad ecological science). By defining clearly what we consider as “post-breeding migration” in our manuscript (see Methods), we believe that we provide the most accurate term for the reader.

Please see also the detailed response in the general comments section about the term “migration”.

Line 93. A 1.5g GLS unit is more than 1% of a 110g individual. I would like to see data on body masses of individuals used presented in the Supplementary Materials at the very least.

Response: erratum, the 110 g individual corresponds to a South-Georgian Diving Petrel, a species that was included in an early version of the manuscript. The sentence was updated accordingly “The mass of the device attached to a plastic or metal ring with a cable tie was < 1.5 g, corresponding to on average $1.07 \pm 0.1\%$ of body mass (117 g – 175 g)”.

Additionally, the mass of each individual was added in the data repository.

Line 107 – there are molt details for Heard I in the primary literature – rather than HANZAB.

Response: the paragraph was restructured and the reference Woehler (1991) was added.

Line 111. I don’t necessarily agree that more time on water reflects molt in CDP (see below).

Response: we agree that a high proportion of time spent on the water could suggest an intense feeding period (which is not incompatible with wing moult period). This point was added in

the Discussion. Please see also our detailed response in the general comments about the wet/dry sensor analysis.

Line 134. Unrealistic distance not speed is >1500km/day.

Response: the unit was updated to 1500km/day.

Lines 135-136. Authors describe beginning of outward and end of inward – what about the end of outward or the start of inward trips? If these birds are “migrating” (which they are not) these dates could be defined. However, as the CDP are dispersing post-breeding rather than migrating, these dates are impossible to define and the kernel analyses reinforces my concerns re confounding of dispersal and migration.

Response: We understand the concerns raised regarding of a potential confounding of dispersal and migration. As suggested, the data about the outward and inward trips were added in table 1.

Additionally, Figure S2 in the supplementary data provides the tracking data of 3 individuals in order to illustrate the migration of CDP, highlighting clear outward and inward movements. Please see also our detailed response in the general comments section about the term “migration”.

Line 150. There is a circularity in the assumption that more time spent sitting on water corresponds with molt, and that molt decreases flying efficiency and therefore the CDP will spend more time sitting on water. The 2 cited studies are for much larger flying birds and it is questionable that the results for albatrosses are true for CDP.

Response: we agree that more time spent on the water can also indicate intense feeding behaviour. Please see our detailed response in the general comments section about the wet/dry sensor analysis.

Nevertheless, the main study cited here (Cherel et al. 2016) was carried out on blue petrel, Antarctic prion and Thin-billed prion, 3 similar size species to diving petrels (see also Jones et al. 2020 focusing on 2 other species of prions and extended literature on small alcid). Although prions and diving petrels don't have the same foraging ecology, the morphologically and ecologically similar alcid species also exhibit a flightless period when they moult their flight feathers (Bridges 2006). A rapid/synchronous wing moult in diving birds was suggested to be a process of avoiding a prolonged moult-induced disruption of flight ability (Bridges 2004), which would be particularly important for such species with high wing loading.

CDP spend plenty of time on the water close to colonies and close to prey patches such as slicks and fronts, so I suspect there is greater behavioral variability in CDP to explain on water as solely limited to molt periods. (and continuing at Lines 236-244. The periods at sea and on water correspond with the Southern Ocean winter – poor weather, especially close to SACCF). The birds may well be sheltering on the surface that trying to fly around in turbulent air, and they are likely to be feeding to deposit body reserves for the coming breeding season. I don't believe they are constrained in flight abilities due to molt. You could speculate but resting/feeding behaviors are equally compelling hypotheses)

Response: The Discussion was modified accordingly. Please see our detailed response in the general comments section about the wet/dry sensor analysis.

Line 167. Replace ‘prior to’ with ‘before’. One has prior knowledge, one does nothing ‘prior to’ anything. Ditto line 381.

Response: “prior” was replaced by “before”.

Line 185. Please justify the selection of Gaussian models, and whether the associated assumptions were met.

Response: the paragraph about the general linear models was restructured in order to provide the information necessary to justify the family selected for the model.

Lines 188 – 192. When were parametric and non-parametric analyses undertaken? Would be useful to describe explicitly for readers.

Response: the details about parametric and non-parametric tests were added in the Methods. Additionally, in the Results section, the details of the tests used are provided for each analysis.

Line 210. SACCF – missing ‘Current’ in text.

Response: “Current” was added.

Lines 213 – 222. The use of atmospheric fronts by burrowing petrels is well known – the birds wait until after the passage of a cold front over the breeding island, then depart on foraging trips or longer – taking advantage of a SW air flow after the passage of the front – see the publications from Marion I.

See also the work on short-tailed shearwaters (Raymond et al, Woehler et al, Schaeffer et al) as examples. I suspect (Line 218) colony departures were linked with breeding phenology AND local weather systems.

The most interesting aspect of Figure 2 is why do Mana Island CDP fly SE rather than SW into prevailing winds? Either wind flow regionally is different and/or the area to where the CDP disperse is more productive than other areas within the same dispersion distance.

Response: Yes, it is very interesting to see that Mana Island CDP went in a different direction than the other populations. This emphasize the theory of migration vs dispersion. Please see our response about the migration/dispersion related comments.

The influence of weather on the behaviour of seabirds is a very interesting point. There are several studies highlighting the use of air flow in order to minimize the cost of flight.

However, most of the literature focus generally on larger species (shearwaters...) during the breeding period. In the present study, for the CDP from Kanowna for example, the departure occurred late spring or early summer (lower probability of cold fronts at this latitude), and for all the individuals, the local atmospheric pressure (and wind direction and speed) was similar at the departure time than 2 days before and 2 days after (indicating no weather change).

Additionally, all the individuals departed on average 1 day after the last burrow attendance (and fed the chick for the last time; Eizenberg 2019). The lack of cold front during this period combined to a rapid departure in migration after the last burrow attendance suggest a strong influence of phenology in departure time.

Line 256. Please remove ‘interestingly’ as subjective. Ditto Line 294 ‘outstanding’. Ditto line 354 ‘profound’. Line 374 ‘stark’.

Response: “interestingly” was removed, “outstanding” was replaced by “long”, “profound” and “stark” were replaced by “important”.

Lines 267 – 287. The question of intra- versus inter-annual variabilities needs further discussion. If intraannual variability is greater than inter-annual, it will be hard to make generalisations about the one year when data are available from all three sites. Are there independent data on productivity that could be used to examine the annual isotope signatures?

How great are the associated latitudinal differences associated with the stated differences in $\Delta^{15}\text{N}$?

Response: we initially included a section in the Discussion about the inter-annual variation in $\Delta^{15}\text{N}$. However, because the Discussion is too long (and still need to be reduced), we removed this section as it was not a key result in the “post-breeding behaviour” of CDP (the results in $\Delta^{15}\text{N}$ are similar to what was found in other studies and match with the current knowledge of the CDP trophic level).

The inter-annual variability in $\Delta^{15}\text{N}$ could be due to a variation in isotopic baseline signatures. Variations in oceanic factors such as Chl-a concentration can substantially alter $\Delta^{15}\text{N}$ independently of any change in diet (Polito et al. 2019).

Line 292 (and 309). Why is wing loading “important”?

Response: “important wing loading” refers to the ratio body weight/wing surface. More information has been added in the Methods, and a reference was added in this sentence.

Line 301. The CDP dispersed TO or AROUND oceanic frontal systems, not in.

Response: “in” was replaced by “to”.

Lines 301 – 307. The inter-island dispersions reflect post-breeding dispersion to nearest high(er) productivity areas in adjacent areas of the Southern Ocean where the CDP could undertake molt and deposit reserves for the next breeding season. Line 307 – “behavioural homogeneity” – the productive areas used are simply within foraging/dispersal capacities of CDP colonies in New Zealand.

Response: Agreed, this could indeed reflect that each population exploits the nearest productive habitats. This point is discussed in the following paragraphs.

Lines 320 – 330. Suggest delete and confine discussion to previous paragraph re oceanic productivity. The text is speculative with no supporting data.

Response: The speculative part of the paragraph was deleted, and the paragraph was restructured.

Line 342. What about founder effects potentially generating population-specific phenologies?

Response: the sentence was rephrased in order to provide more information about this point. Nevertheless, although founder effects could potentially have affected the phenological differences among populations, we would need more information about the population genetic. In the present study, the important inter-annual variability in the phenology within population (in particular Kanowna Island) highlights that the CDP has the capacity to adapt its phenology quickly in relation to the local oceanic conditions (see Eizenberg 2019).

Line 385. Are there any independent data to suggest a decrease in marine productivity?

Response: yes, Moore & Abbott (2002) described a peak of productivity near the PF in December, followed by a rapid decrease the following months. This reference is cited in the sentence.

Line 399. Nocturnal vocalisations at colonies are both territorial activity and also displaying/soliciting mates.

Response: agreed. The sentence was modified in order to include both reasons related to nocturnal vocalisations.

Lines 405 – 409 re *Nyctiphanes*. The inter-annual variability in *Nyctiphanes* abundance is driven by regional winds driven by ENSO events – see Mills et al ref #78. Kerguelen, Kanowna and South Georgia etc CDP populations all influenced by ENSO and/or ACW and/or other dipole oceanographic/atmospheric phenomena.

Response: Yes, ENSO influences the marine ecosystem in all these regions. However, dramatic variations in zooplankton community as we can see in south-east Australia and NZ are not common in sub-Antarctic regions. See Evans et al. (2020) about the impact of marine heat wave in south-eastern Australia zooplankton community. See also Eizenberg (2019) about the unusual variation in phenology and breeding success of CDP in Bass Strait.

Lines 383 – 412 are weak and I suggest removing.

Response: We believe that this section provides key elements about the difference among the 3 study populations.

We reduced and restructured the paragraphs in order to clear out what is the main information in this section.

Line 419 – As I understand it, the study looked at breeding CDP? What about dispersal of non-breeding CDB between fledging and initial breeding effort? These birds will not be constrained to a colony and could potentially disperse (not migrate) farther.

Response: Agreed. Discussion about the need to study the spatial distribution of juveniles was added in this section.

Line 420. The study has not “provided clear evidence of segregation of ... CDP populations”. The study has provided clear evidence of segregation of widely-separated CDP populations – which is not the same thing. Useful and interesting results would emerge from studies on closer-spaced populations – if they behave similarly (as you might expect), then that shows some homogeneity. If not, the story could explore competition etc.

Response: the sentence was rephrased to match with the results provided in the manuscript “the present study has provided clear evidence of segregation in the post-breeding migratory destinations of eastern CDP populations.”

As suggested, it would be very interesting to investigate the non-breeding distribution and the genetic structure within the Indian Ocean basin (Marion/Crozet/Kerguelen/Heard).

Nonetheless, regarding that strong genetic isolation at a small spatial scale was found for Peruvian diving petrels (< 300km), we can expect similar results for these populations (between island distance varies from 500km to 1,300km). Additionally, if the dispersion of juvenile matches the direction of migration of adult birds (heading south-west), there is little chance for a bird to reach any neighbour island.

Figure 3. Presumably these are for breeding adults, so the caption should state this explicitly. Why are year-round kernels smaller and substantially shifted compared to the winter kernels?

Response: the sentence “The devices were deployed on breeding birds” was added in the caption of Figure 3 and 4.

I am not sure I understand the comment about the difference between year-round kernels and “winter” kernel. The year-round kernels (Figure 3, year-round distribution divided in 4 periods of 3 months) were estimated based on all the locations for each period of 3 months. In Figure 4, the post-breeding distribution was estimated based on the location between the departure and the return of each birds (see Methods section for more details). Additionally, we took care to not use the term “winter” migration/distribution because this term is accurate only for Kerguelen population. In Australia and New Zealand, the birds are in the vicinity of their colony during this period (the migration is in summer).

Supplementary Figures S1 – S5. Please state sample sizes n explicitly in all figure legends.

Response: all the figure legends were updated.

Appendix C

Response to the editor

Thank you for your positive feedback.

All the minor revisions were added in the last version of the manuscript.